# PMIP4/CMIP6 Last Interglacial simulations using three different versions of MIROC: importance of vegetation

Ryouta O'ishi[1], Wing-Le Chan[1], Ayako Abe-Ouchi[123], Sam Sherriff-Tadano[1], Rumi Ohgaito[3] and Masakazu Yoshimori[1]

[1]Atmosphere and Ocean Research Institute, University of Tokyo, Kashiwa, 2778568, Japan
[2]National Institute of Polar Research, Tachikawa, 1908518, Japan
[3]Japan Agency for Marine-Earth Science and Technology, Yokohama, 2360001, Japan

*Correspondence to*: Ryouta O'ishi (ryo@aori.u-tokyo.ac.jp)

**Abstract.** We carry out three sets of Last Interglacial (LIG) experiments, named lig127k, and of Pre-Industrial experiments, named piControl, both as part of PMIP4/CMIP6, using three versions of the MIROC model: MIROC4m, MIROC4m-LPJ and MIROC-ES2L. The results are compared with reconstructions from climate proxy data. All models show summer warming over northern high latitude land, reflecting the differences between the distributions of the LIG and present-day solar irradiance. Globally averaged temperature changes are -0.94K (MIROC4m), -0.39K (MIROC4m-LPJ) and -0.43K (MIROC-ES2L). Only MIROC4m-LPJ, which includes dynamical vegetation feedback from the change in vegetation distribution, shows annual mean warming signals at northern high latitudes, as indicated by proxy data. In contrast, the latest Earth System Model (ESM) of MIROC, MIROC-ES2L, which consider only a partial vegetation effect through the leaf area index, shows no change or even annual cooling over large parts of the northern hemisphere. Results from the series of experiments show that the inclusion of full vegetation feedback is necessary for the reproduction of the strong annual warming over land at northern high latitudes. The LIG experimental results show that the warming predicted by models is still underestimated, even with dynamical vegetation, compared to reconstructions from proxy data, suggesting that further investigation and improvement to the climate feedback mechanism are needed.

## 1 Introduction

The Last Interglacial (LIG, 130ka-116ka) is referred to as the warmest period in the recent glacial-interglacial cycle (NGRIP members 2004, Overpeck et al. 2006, Lang and Wolff 2011). The most important characteristic of the LIG is the strong summer solar irradiance in the northern hemisphere due to the difference in the earth's orbit during that period and that of the present-day (Berger 1978, Yin and Berger 2015). For example, 127,000 years ago, the peak summer solar irradiance was more than 70W/m$^2$ larger compared to that of the present-day at 65° N (Figure 1). Geological evidence shows that the globally averaged temperature is higher by 1.9K in the LIG compared to the pre-industrial (Turney and Jones 2010). The sea surface temperature (SST) also shows a warming of 0.7±0.6K (McKay et al. 2011) and 0.5±0.3K (Hoffman et al. 2017). Records from ice cores show warming in Greenland (NEEM 2013, Masson-Delmotte et al. 2011) and in Antarctica (Jouzel et al. 2007, Stenni et al.

2010, Capron et al. 2017). Sea level rise due to warming has also been pointed out, with contributions from the mass balance change in the Greenland and the Antarctic ice sheets. The total sea level rise in the LIG is estimated to be 5.5-9.0m (Dutton and Lambeck 2012, Dutton et al. 2015). The contribution from the Greenland ice sheet is estimated to be 1.4-4.3m (Robinson et al. 2011, Born and Nisancioglu, 2012, Quiquet et al. 2013, IPCC 2013), 2-4m (Dutton and Lambeck 2012) and 0.6-3.5m (Stone et al. 2013). Paleo-evidence shows substantial summer warming at northern high latitude land areas (typically +4-5K, at most +8K) in response to this different spatial and temporal pattern of solar irradiance (CLIP members 2006, Otto-Bliesner et al. 2006, Velichko et al. 2008). Northward shift of boreal treeline due to warming is also indicated by proxies (LIGA members 1991, Edwards et al. 2003). In the Sahara, vegetation is estimated to be expanded to northward due to wetter condition (Larrasoaña et al. 2013).

The Paleoclimate Modelling Intercomparison Project (PMIP) coordinates the cooperation and comparison between modelling and data of the past (Braconnot et al. (2000, 2007, 2012), Kageyama et al. 2018). The LIG is one of the targeted periods in addition to the mid-Holocene and the Last Glacial Maximum (Lunt et al. 2013, Otto-Bliesner et al. 2017). For the simulation of the past periods, PMIP provides protocols with common settings which should be applied to participating models. In the present study, we apply the LIG boundary conditions provided by the PMIP4 to three different versions of atmosphere-ocean coupled general circulation models (AOGCMs) which belong to the Model for Interdisciplinary Research on Climate (MIROC) family and compare results with the pre-industrial simulations, focusing on the different treatment of vegetation among the three models. In section 2, models and their components are described. The experimental settings are also mentioned in section 2. Section3 describes the results of LIG model simulations and comparison with proxies. In section 4, we discuss the validity of data-model comparison. The feedback mechanism for the Arctic warming amplification is also discussed.

## 2 Models and settings

### 2.1 Models

In this section, models used in the present study, MIROC4m, MIROC4m-LPJ and MIROC-ES2L, are described. An overview of the models is shown in Figure 2. Their critical difference among others is that MIROC4m prescribes modern vegetation distribution with corresponding leaf area indices, MIROC-ES2L prescribes modern vegetation distribution but allows for leaf area indices to respond to the simulated climate, and MIROC4m-LPJ simulates both vegetation distribution and leaf area indices in response to the climate change. Therefore, comparisons of these three models provide an opportunity of loosely revealing the effect of no, partial, and full vegetation feedbacks. We note, however, that the effect of other differences in model resolution and physics cannot of course be excluded. MIROC4m and MIROC-LPJ do not include carbon feedback upon the climate. MIROC4m-LPJ predicts change of land surface properties depending on vegetation change. MIROC-ES2L includes prediction of atmospheric carbon and nitrogen content as greenhouse gases (GHGs), but the radiative effect of GHGs is fixed in the present study (see settings).

### 2.1.1 MIROC4m

The AOGCM, MIROC4m, is based on MIROC3.2 which contributed to the fourth assessment report (AR4; Meehl et al. 2007) of the Intergovernmental Panel on Climate Change (IPCC). MIROC4m consists of Center for Climate System Research Atmosphere General Circulation Model (CCSR AGCM; Hasumi and Emori 2004) and CCSR Ocean Component Model (COCO; Hasumi 2006). The AGCM solves the primitive equations on a sphere using a spectral method (Numaguti et al. 1997). The model resolution of the atmosphere component is T42 with 20 vertical layers. The land submodel is the Minimal Advanced Treatments of Surface Interaction and Runoff (MATSIRO; Takata et al. 2003). Vegetation is prescribed as a satellite based present-day distribution. The OGCM solves the primitive equation on a sphere, where the Boussinesq and hydrostatic approximations are adopted (Hasumi 2006). The horizontal resolution is ~1.4° in longitude and 0.56°–1.4° in latitude (latitudinal resolution is finer near the equator), and there are 43 vertical layers. The OGCM is coupled to a dynamic–thermodynamic sea ice model (Hasumi 2006). MIROC4m has been used extensively for modern climate (Obase et al. 2017), paleoclimate (Ohgaito and Abe-Ouchi 2007; Abe-Ouchi et al. 2013; Sherriff-Tadano et al. 2018), and future climate studies (Yamamoto et al. 2015).

### 2.1.2 MIROC4m-LPJ

We recently developed a vegetation coupled AOGCM MIROC4m-LPJ by introducing the Lund-Potsdam-Jenna Dynamical Global Vegetation Model (LPJ-DGVM; Sitch et al. 2003) into MIROC4m. The coupling method is similar to that used for MIROC-LPJ in previous studies (O'ishi and Abe-Ouchi 2009, 2011, 2013) which adopted a motionless 'slab' ocean instead of the dynamical ocean model COCO. What is new in the present study is that bias correction is not applied to the coupling between atmosphere and vegetation components. LPJ-DGVM predicts potential vegetation distribution which is translated to the classification used for MATSIRO annually by using monthly mean temperature, precipitation and cloud cover predicted by the atmosphere component of the GCM. LPJ-DGVM predicts vegetation distribution based on carbon balance, but MIROC-LPJ does not include a carbon cycle or its feedback to the climate. Leaf area index (LAI) seasonality in MIROC-LPJ is prescribed with a sine curve which is defined by maximum and minimum values for each vegetation type. A detailed description of the method can be found in O'ishi and Abe-Ouchi (2009). Another important modification is the introduction of a wetland scheme developed by Nitta et al. (2017). This scheme improves seasonality of the hydrological behaviour over land. When snowmelt occurs, part of the meltwater does not immediately discharge into rivers but remains in an isolated storage. This stored water decays with a timescale dependent on the standard deviation of the topography and the amount of decayed water is taken into account in the land surface water and energy balance. The introduction of this scheme reduces summer warm bias over land at mid to high latitudes. The model resolutions of the atmosphere and land are the same as those of MIROC4m. The resolution of LPJ-DGVM is T42.

### 2.1.3 MIROC-ES2L

An earth system model (ESM) MIROC, Earth System version 2 for long-term simulations (MIROC-ES2L; Hajima et al. 2020, Ohgaito et al. 2020) is one of the contributing models to PMIP4/CMIP6. The physical component of MIROC-ES2L is MIROC5.2 (Tatebe et al. 2018), an upgraded version of MIROC5 (Watanabe et al. 2010) which contributed to the IPCC AR5 (IPCC, 2013). In MIROC-ES2L, the sea ice component is updated from MIROC4m to a sub-grid multicategory model described in Komuro and Suzuki (2013). The most important update in MIROC-ES2L from previous versions of MIROC-ESM (Watanabe et al. 2011, Kawamiya et al. 2005) is the introduction of a nitrogen cycle. The land nitrogen and carbon cycles are predicted by a modified version of the Vegetation Integrative Simulator for a Trace gas model, VISIT (Ito and Inatomi, 2012a), referred to hereafter as VISIT-e. Vegetation distribution is prescribed in both MATSIRO and VISIT-e, but the LAI is predicted daily by VISIT-e and transferred to MATSIRO. The ocean nitrogen and carbon cycles are predicted by an ocean biogeochemical component model OECO2. Detailed information is described in Hajima et al. (2020) and Ohgaito et al. (2020). The model resolution of the atmosphere component is T42 with 40 vertical layers. The model resolution of the ocean component is a warped tripolar coordinate system with longitudinal 1° grid spacing in the spherical coordinate south of 63°N and meridional grid spacing varying from 0.5° (near the equator) to 1° (mid-latitudes). The number of vertical layers is set to 63.

## 2.2 Settings

In the present study, three models are run with the same forcings and with the boundary conditions of the pre-industrial (*piControl* in PMIP4/CMIP6, hereafter PI) and of the Last Interglacial (*lig127k* in PMIP4, hereafter 127k), as shown in Table 1, following the PMIP4/CMIP6 protocol (Otto-Bliesner et al. 2017). The orbital forcings in both experiments are the same as those recommended in Otto-Bliesner et al. (2017). The GHG concentrations in *piControl* are slightly different from Otto-Bliesner et al. (2017). We apply fixed GHG values from the CMIP3 control experiment to MIROC4m and MIROC4m-LPJ. In MIROC-ES2L, GHGs are fixed to the same as the CMIP6 DECK *piControl* experiment settings. The GHG concentrations in the 127k experiments are fixed to the same as those specified in Otto-Bliesner et al. (2017) in all GCMs. Details of these GHG values are shown in Table 1. Especially in MIROC-ES2L, carbon balance in the land and ocean ecosystem does not affect atmospheric $CO_2$ concentration in the present study. Paleogeography and ice sheet are set to modern in all experiments. Vegetation distribution in MIROC4m is fixed to present-day configuration according to Ramankutty and Foley (1999); see Figure 3. In MIROC4m-LPJ, vegetation distribution is predicted as in MIROC-LPJ (O'ishi and Abe-Ouchi 2006). MIROC-ES2L applies a new definition of the land-sea mask, different to that of MIROC4m. The distribution of prescribed vegetation is also updated from MIROC4m and re-defined by using newer satellite data sets. In MIROC-ES2L, vegetation distribution is fixed to a satellite-based vegetation distribution (Matthews 1983, Matthews 1984, Hall et al. 2006), the same as that in MATSIRO of MIROC5 (Watanabe et al. 2010) (see Figure 2) and in the DECK *piControl* experiment (Hajima et al. 2020, Ohgaito et al. 2020) using VISIT-e. However, as described above, VISIT-e predicts the LAI which is accessed by MATSIRO. In PI experiments, MIROC4m, MIROC4m-LPJ and MIROC-ES2L are integrated for 4000 years, 2000 years and 500 years, respectively. In 127k experiments, MIROC4m, MIROC4m-LPJ and MIROC-ES2L are integrated for 3000 years, 3000 years

and 1550 years, respectively. In all experiments, the last 100 years, during which the climate has reached equilibrium, are used for analysis. Since the definition of length of months is set to the present-day calendar, monthly averaged values may not correspond exactly to the appropriate seasons in 127ka. We applied a calendar adjustment on all 127k results based on Bartlein and Shafer (2019). Their method defines the length of months based on the arc of the Earth's orbit which has been traverses.

## 3 Results

### 3.1 Temperature

A look at the globally averaged annual mean temperature changes in the present study shows that there is a slight cooling from PI for all three models: -0.94K(MIROC4m), -0.39K(MIROC4m-LPJ) and -0.43K(MIROC-ES2L). Seasonally and annually averaged surface air temperature differences between 127k and PI are shown in Figure 4. All models show the largest regional warming (> 6K) in June-July-August (JJA) over northern high latitude land and largest global cooling in December-January-February (DJF) which corresponds to increased boreal summer solar irradiance and decreased boreal winter solar irradiance in the LIG, respectively. In September-October-November (SON), there is still a slight warming over northern hemisphere land at mid and high latitudes carried over from the summer. In March-April-May (MAM), the global cooling is less than that in DJF and warming is seen at northern high latitudes. Antarctica in all models show cooling in DJF, reflecting the decrease of solar irradiance at SH high latitudes in winter (Figure 1).

The change in annually averaged surface air temperature is smaller than that of the seasonally averaged value because summer warming is compensated by cooling in other seasons. MIROC4m shows global cooling except for some isolated areas where annual temperatures are higher, e.g. Greenland. MIROC4m-LPJ shows annual warming (at most 3K) both over high latitude land and in the Arctic Ocean with vegetation changing from tundra to boreal forest in the northern coastal areas of Eurasia and North America (Figure 3). Only MIROC4m-LPJ shows strong warming in Alaska and Eastern Siberia, especially in MAM. The Arctic Ocean shows warming throughout all seasons in MIROC4m-LPJ, which is not seen in MIROC4m and MIROC-ES2L. MIROC-ES2L, as well as MIROC4m-LPJ, show annual warming in the Arctic Ocean, but the intensity in the former model is less. MIROC-ES2L shows annual warming in Antarctica which is not seen in the other two models.

May-June-July-August (MJJA) averaged temperature difference is shown in Figure 5, which focuses on Greenland, because the ice sheet mass balance is affected by the increased solar irradiance during MJJA at northern high latitudes (Figure 1). MIROC4m and MIROC4m-LPJ show a similar pattern of warming which increases with altitude. MIROC-ES2L shows a more homogeneous warming pattern over Greenland and the intensity of warming is weaker than in the other two models.

Simulated temperature changes are compared with reconstructed values from proxies. In Figure 6, annual surface temperature change is compared with land proxies by Turney and Jones (2010), hereafter referred to as TJ2010. MIROC4m-LPJ exhibits the largest warming among the three models at northern high latitudes. However, the agreement with TJ2010 is not quantitative but more qualitative; annual warming is still underestimated. The other two models show cooling over northern high latitude land. Greenland warming appears in all models, but is smaller than that of TJ2010. In Antarctica, MIROC-ES2L shows the same sign of temperature change as in TJ2010, but the intensity of warming (at most 1K) is weaker than that of TJ2010.

MIROC4m and MIROC4m-LPJ show cooling rather than warming in Antarctica. In Figure 6, annual surface temperature change is also compared with a newer reconstruction by Capron et al. (2017), hereafter referred to as C2017. All models show warming in Greenland, but only MIROC-ES2L reproduces Antarctica warming. The models underestimate warming at all sites in C2017 as the intensity is not reproduced.

Figure 7 compares the simulated annual sea surface temperature (SST) change and TJ2010 ocean proxies. All three models predict warm SST at northern high latitudes and cooling in tropical regions. This large-scale latitudinal pattern agrees with TJ2010, but some individual sites disagree in terms of sign. For example, the NH SST warming is more substantial in MIROC4m-LPJ and in MIROC-ES2L than in MIROC4m. The largest warming in the SH is seen in MIROC-ES2L. However, the intensity of SST changes in all models is far smaller than that of TJ2010. We also compare modelled annual SST difference

with newer reconstructions by C2017 and Hoffman et al (2017), hereafter referred to as H2017, in Figure 7. MIROC4m underestimates warming in the Irminger Sea and shows changes of opposite sign in the southern part of the Pacific and Indian Oceans. MIROC4m-LPJ shows larger NH warming than MIROC4m does. However, as with MIROC4m, warming in SH is not simulated. MIROC-ES2L predicts better warming in the Irminger Sea than MIROC4m/MIROC4m-LPJ does. MIROC-ES2L also predicts improved warming in SH which is partially consistent with proxies. Summer temperature change in the

models are compared with that of reconstructions by C2017 and H2017 in Figure 8. Across the wide expanse of the northern Atlantic Ocean, all models predict warming whose sign is consistent with that of reconstructions, except at some sites which show cooling. MIROC4m-LPJ predicts the largest warming (at most > 4K) while the other two models show a smaller intensity of warming (at most 3K) in the northern Atlantic Ocean. On the other hand, in SH, MIROC4m and MIROC4m-LPJ show cooling in contrast to the warming indicated by proxies. MIROC-ES2L show warming across much of the Southern Ocean, in

contrast to the other two models. However, some sites still indicate an opposite sign to that of proxies.

## 3.2 Precipitation, sea ice and vegetation

### 3.2.1 Precipitation

The precipitation change between 127k and PI is shown in Figure 9. In general, the annually averaged precipitation changes

seen in all three models are similar and show the largest increase, 700mm/year, at the southern edge of the Sahara Desert. The second largest increase over land is seen in the northern part of India. The largest precipitation increase over ocean is seen in the Somali Sea. On the other hand, precipitation decrease is mainly seen in the southern hemisphere. South America, South Africa and the northern part of Australia become drier than PI. These characteristics in the model result are consistent with previous studies (Scussolini et al. 2019), except for the slight annual precipitation increase in eastern Siberia and Alaska. In

our three MIROC versions, precipitation increase in eastern Siberia and Alaska is only seen in JJA. MIROC4m and MIROC4m-LPJ show almost the same precipitation change since they share the same atmosphere component. MIROC-ES2L shows a pattern slightly different to that of the MIROC4m models.

### 3.2.2 Sea ice

In Figure 10, March NH sea ice concentration in PI, 127k and their difference (127k-PI) are shown for all three models. The PI sea ice distribution show characteristics common in both MIROC4m and MIROC4m-LPJ. In PI, they show values larger than observation in the Lablador Sea, the Irminger Sea and the Beaufort Sea. MIROC-ES2L shows a pattern different to the two MIROC4m models. The sea ice concentration in MIROC-ES2L is more realistic than MIROC4m and MIROC4m-LPJ. This is due to the different ocean and sea ice model adopted in the physical part of MIROC-ES2L. However, sea ice
concentration in 127k and PI during March do not show any clear differences. Figure 11 shows September NH sea ice concentration. As in March, MIROC-ES2L shows a different distribution of sea ice and response in 127k compared to MIROC4m-based models. MIROC4m-LPJ shows the largest reduction in sea ice from PI and 127k which corresponds to a warm Arctic Ocean during those two periods. As such, MIROC4m-LPJ predicts less sea ice in September in PI compared to observation (e.g. HadISST averaged over 1870-1919; Rayner et al. 2003). March SH sea ice concentration is shown in Figure
12. In the Southern Ocean, all models show sea ice extent in PI to be smaller than observation. March sea ice shows the same characteristics in the two MIROC4m-based models. MIROC-ES2L shows a different pattern, and the smallest amount of sea ice, compared with MIROC4m-based models. In all three models, March sea ice increases in 127k but the amount differs depending on the model, similar to September NH. September SH sea ice concentration is shown in Figure 13. As in March, there is a discrepancy between the sea ice concentrations in the MIROC4m-based models and in MIROC-ES2L. Sea ice in the
PI is smaller in MIROC-ES2L and the response of sea ice in 127k is also the smallest in that model. MIROC-ES2L clearly underestimates sea ice in both seasons compared to observation.

### 3.2.3 Vegetation

We compare the vegetation distribution in all three models (Figure 3) regardless of different treatment (prescribed or predicted).
MIROC4m and MIROCES2L adopt a fixed vegetation distribution based on satellite data. Vegetation distribution in MIROC4m is based on the classification of MATSIRO, translated from actual vegetation by Ramankutty and Foley (1999). MIROC-ES2L is also fixed to satellite-based vegetation distribution which is translated from satellite data (Matthews 1983, Matthews 1984, Hall et al. 2006). These two vegetation maps show similar patterns of forest and grassland, although differ in the interpretation of classification such as C3/C4 or the boundary between forest and tundra.  Only MIROC4m-LPJ predicts
vegetation distribution in the present study. The 100-year averaged vegetation distribution in the PI shows characteristics common with the other two satellite-based distributions, except for the overestimation of forests (in boreal forest band and African tropical forest) and underestimation of grassland (in African Savanna and central Eurasia). In the 127k simulation, vegetation changes drastically at northern high latitudes. Tundra is broadly replaced by boreal deciduous forest and almost disappears, reflecting the summer warming in the northern high latitude land especially in Eastern Siberia and North America.
Forestation of tundra regions causes amplification of warming in Eastern Siberia and North America, especially in the snow melting season. This northward shift of boreal forest eventually leads to an increase in the annually averaged temperature of Eastern Siberia and North America by an additional 3K compared with LIG warming without vegetation change. Grassland appears over a wide area at the boreal-temperate boundary in both Eurasia and North America due to less precipitation

supporting forest growth in 127k. This increase in grassland causes cooling at mid-latitudes, especially in Eurasia. In the Sahara,
a slight northward expansion of grassland is seen, but MIROC4m-LPJ does not reproduce the so-called "green Sahara" (Larrasoaña et al. 2013). Figure 14 shows JJA zonally averaged precipitation over the Sahara (30W-20E). In all models, 127k summer precipitation shifts northward compared to PI. MIROC4m-LPJ shows the largest northward shift of precipitation.

## 4 Conclusions and Discussions

In the present study, we examined the LIG and PI simulations in accordance to the PMIP4 protocol by using three different
versions of the MIROC AOGCM. These three models show a similar response of temperature to the LIG boundary conditions, i.e. warming in boreal summer and cooling in boreal winter (Figure 4). However, the annually averaged temperature is different among the models. Only MIROC4m-LPJ predicts annual warming at NH high latitudes qualitatively consistent with proxy data such as Turney and Jones (2010), Capron et al. (2017) and Hoffman et al. (2017), while the other two models show a cooling at NH high latitudes. Capron et al. (2014, 2017) noted that proxies in Turney and Jones (2010) were based on peak
warmth values throughout the LIG, and thus the 127k result would not be directly comparable with Turney and Jones (2010). Although it would be more appropriate to compare modelled LIG result with time slice proxy data at 127ka from time series reconstructions (Capron et al. 2017, Hoffman et al. 2017), Turney and Jones (2010) provides a large-scale pattern of temperature change. Hence comparisons between LIG simulation and Turney and Jones (2010) would still be of much value.

The vegetation change seen in MIROC4m-LPJ simulations is a reasonable response to temperature change induced by
modifications in orbital parameters. The largest change is the northward shift of boreal forest and expansion of grassland at mid-latitudes. By comparing MIROC4m-LPJ and MIROC4m, we suggest that the vegetation feedback mechanism is necessary to explain the temperature change reconstructed by proxies since MIROC4m-LPJ predicts warming closer to reconstructions. By considering the overestimation of boreal forest in PI, vegetation feedback may still be underestimated in MIROC4m-LPJ. The introduction of dynamical vegetation in MIROC4m-LPJ appears to amplify the warming not only over land but also in
the ocean at NH high latitudes. On the other hand, MIROC-ES2L, which partially introduces a vegetation effect through LAI prediction, does not show enough warming in LIG, and even shows annual cooling over land at northern high latitudes. In the Arctic Ocean, all three models show warming in SON, in spite of less solar irradiance at 127k. This can be considered to be the same as an interseasonal effect of warming in the Arctic Ocean shown by Laîné et al. (2016) and Yoshimori and Suzuki (2019). They analysed the energy balance and concluded that heat is stored in the Arctic Ocean during summer and emitted in
autumn and winter which causes a larger warming in autumn and winter than in summer. This commonly occurs in $CO_2$-induced cases (Laîné et al. 2016) and orbitally-induced cases (Yoshimori and Suzuki 2019). In the present study, this mechanism occurs in MIROC4m, without vegetation feedback, and is amplified in MIROC4m-LPJ by vegetation feedback. The largest land surface albedo change occurs in spring (Figure 15) caused by surface-albedo feedback due to the snow-masking effect of trees in MIROC4m-LPJ. The additional energy input to land due to the reduction of albedo is transferred to
the Arctic Ocean, where it reduces sea ice and is stored as heat in the ocean. This leads to largest Arctic warming occurring

in autumn through heat release from the ocean in all three models. To confirm this mechanism, we applied the same feedback analysis method as Yoshimori and Suzuki (2019) and obtained the monthly decomposed contribution of energy flux terms to the surface temperature change between 127k and PI in northern high latitude land and ocean (Figure 16). The result shows strong land albedo effect in April and May with MIROC4m-LPJ. The corresponding ocean heat uptake is largest in MIROC4m-LPJ. In autumn and winter, heat release from the ocean. These results indicate that vegetation feedback, including changes in vegetation distribution, is necessary to predict past warm climate and such results have implications for future climate simulations. Compared to observation, MIROC-ES2L shows the most realistic PI distribution of sea ice in the Arctic Ocean, owing to a new ice physics model. MIROC4m-LPJ predicts the smallest amount of sea ice in PI among the three models in both March and September, because temperature in MIROC4m-LPJ is generally higher than that of MIROC4m over land due to the inclusion of dynamical vegetation. This higher temperature reduces sea ice in the Arctic Ocean in the PI and thus inevitably affects the response of sea ice to higher temperatures in LIG. On the other hand, in the Southern Ocean, MIROC-ES2L underestimates sea ice extension both in March and September which leads to underestimation of feedbacks related to sea ice. To investigate the mechanisms in detail, we are planning further feedback analysis focusing on surface energy balance.

## 5 Code and Data availability

The codes for MIROC-ES2L, MIROC4m, MIROC4m-LPJ are not publicly archived because of the copyright policy of the MIROC community. Readers are requested to contact the corresponding author if they wish to validate the model configurations of MIROC family models and conduct replication experiments. The source codes, required input data, and simulation results will be provided by the modelling community to which the author belongs. The output of the piControl and lig127k from MIROC-ES2L will be distributed and made freely available through the Earth System Grid Federation (ESGF). Details on the ESGF can be found on the website of the CMIP Panel (https://www.wcrp-climate.org/wgcm-cmip/wgcm-cmip6). All experiments performed with MIROC4m and MIROC4m-LPJ will be available from the FTP server ftp://157.82.240.174/~ryo/cp-2019-172/ .

## 6 Author contribution

All authors contributed to the writing of the paper. RO[1] wrote the draft and carried out the analysis. The model simulations were carried out by RO[1] and WLC. RO[3] made arrangements and supported the simulation. RO[1], WLC, RO[3], SST, MY and AAO participated in discussion. AAO coordinated the study.

## 7 Competing interests

The authors declare no competing interests.

**Acknowledgement**

This work was partially carried out within the Arctic Challenge for Sustainability (ArCS) Project (Grant Number JPMXD1300000000) and Arctic Challenge for Sustainability II (ArCS II) Project (Grant Number JPMXD1420318865) from the Ministry of Education, Culture, Sports, Science and Technology (MEXT), Japan. This work is partly supported by the Integrated Research Program for Advancing Climate Models (TOUGOU programme; Grant Number JPMXD0717935715; MEXT), KAKENHI (Grant 17H06104; JSPS) and KAKENHI (Grant 17H06323; MEXT). The simulations were conducted on the Earth Simulator of JAMSTEC. MY benefited from discussions with Ms. M. Suzuki on vegetation feedback with a slab-ocean coupled atmospheric GCM.

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

|  | PI | 127k |
|---|---|---|
| Eccentricity | 0.01672 | 0.039378 |
| Obliquity (degrees) | 23.45 | 24.04 |
| Perihelion – 180 (degrees) | 102.04 | 275.41 |
| Carbon dioxide (ppm) | 285.431 / 284.725 | 275 |
| Methane (ppb) | 863.303 / 808.249 | 685 |
| Nitrous oxide (ppb) | 279.266 / 273.021 | 255 |
| Solar constant (W/m$^2$) | 1361 | Same as in *piControl* |
| Paleogeography | Modern | Same as in PI |
| Ice sheets | Modern | Same as in PI |
| Vegetation | Prescribed or interactive | Prescribed as in PI or interactive (depends on model) |

Table 1: Forcings and boundary conditions of Pre-Industrial (PI) and the Last Interglacial (127k). Greenhouse gases levels for MIROC-ES2L are shown after / in the PI column.


| Experiment | Model name | Model type | Vegetation | Int. length |
|---|---|---|---|---|
| PI | MIROC4m | AOGCM | Prescribed | 4000 yrs |
| PI | MIROC4m-LPJ | AOVGCM | Interactive | 2000 yrs |
| PI | MIROC-ES2L | ESM | Prescribed | 500 yrs |
| 127k | MIROC4m | AOGCM | Prescribed | 3000 yrs |
| 127k | MIROC4m-LPJ | AOVGCM | Interactive | 3000 yrs |
| 127k | MIROC-ES2L | ESM | Prescribed | 1550 yrs |

Table 2: List of experiments

| No. | Symbol | Physical meaning |
|---|---|---|
| 1 | S-B | nonlinearity of Stefan–Boltzmann law |
| 2 | alb | surface albedo change |
| 3 | alb*clr_sw | nonlinear effect of surface albedo and clear-sky shortwave radiation changes |
| 4 | clr_sw | clear-sky shortwave radiation change |
| 5 | clr_lw | clear-sky longwave radiation change |
| 6 | cld_sw | shortwave cloud radiative effect |
| 7 | cld_lw | longwave cloud radiative effect |
| 8 | evap | surface latent heat flux via evaporation |
| 9 | sens | surface sensible heat flux |
| 10 | surface | net surface energy flux including latent heat for snow/ice melting and heat exchange with the subsurface |
| 11 | synergy | synergy term for local feedbacks and local warming sensitivity |

Table 3: List of the energy flux terms used in Figure 16. Row number 1 represents the strength of the global mean feedback calculated with local warming sensitivity(reproduced from Yoshimori and Suzuki, 2019, Table 3). Row numbers 2–10 represent the strength of local feedback calculated with global mean warming sensitivity.

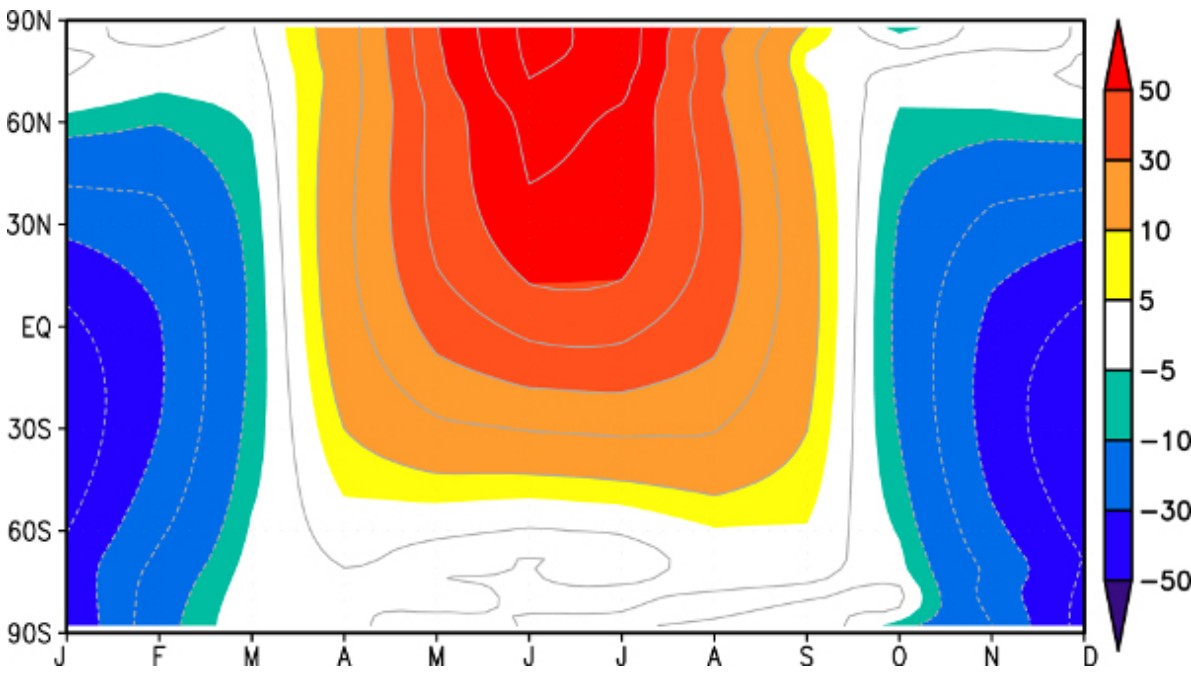

Figure 1: Latitude-month insolation anomaly between 127k and PI (W/m$^2$). Calendar in 127k is adjusted by a method based on Bartlein and Shafer (2019).


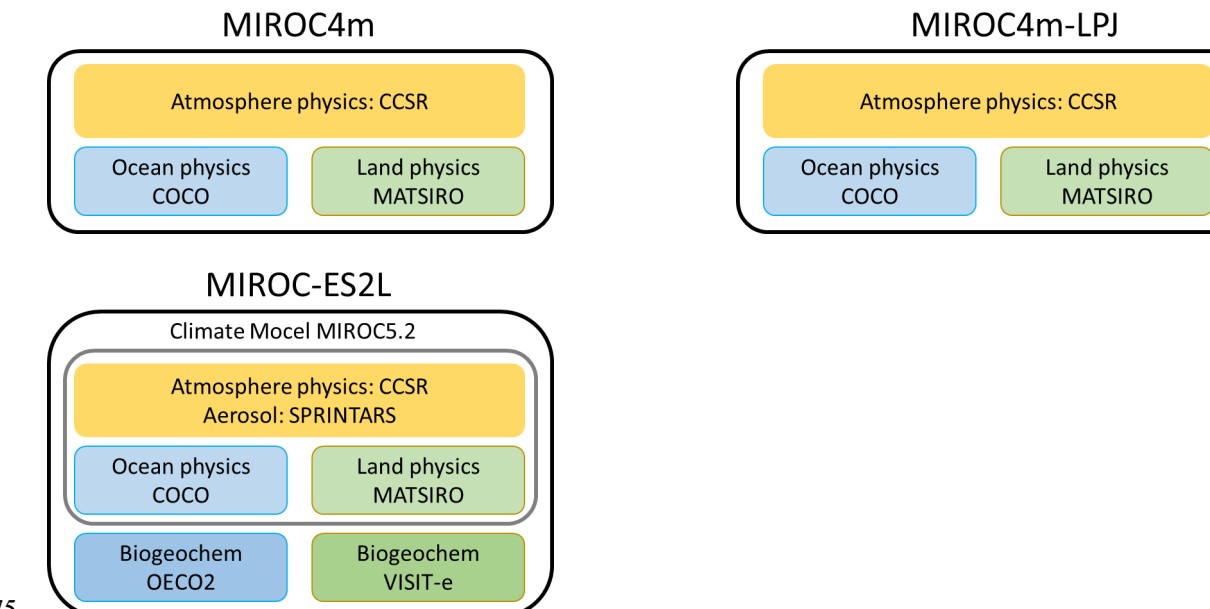

MIROC4m

Atmosphere physics: CCSR

Ocean physics
COCO

Land physics
MATSIRO

MIROC4m-LPJ

Atmosphere physics: CCSR

Ocean physics
COCO

Land physics
MATSIRO

Vegetation
LPJ-DGVM

MIROC-ES2L

Climate Mocel MIROC5.2

Atmosphere physics: CCSR
Aerosol: SPRINTARS

Ocean physics
COCO

Land physics
MATSIRO

Biogeochem
OECO2

Biogeochem
VISIT-e


Figure 2: schematic of models


MIROC4m prescribed

MIROC4-ES2L prescribed

MIROC4m-LPJ Pre-Industrial

MIROC4m-LPJ 127ka

Vegetation

- Ice
- Tundra
- Boreal deciduous forest
- Boreal conifer forest
- Desert
- Grassland
- Savanna
- Temperate forest
- Tropical deciduous forest
- Tropical evergreen forest

Figure 3: Vegetation distribution as fixed boundary condition (MIROC4m and MIROC-ES2L) and resultant most dominant vegetation types in MIROC4m-LPJ experiments in PI and 127k.

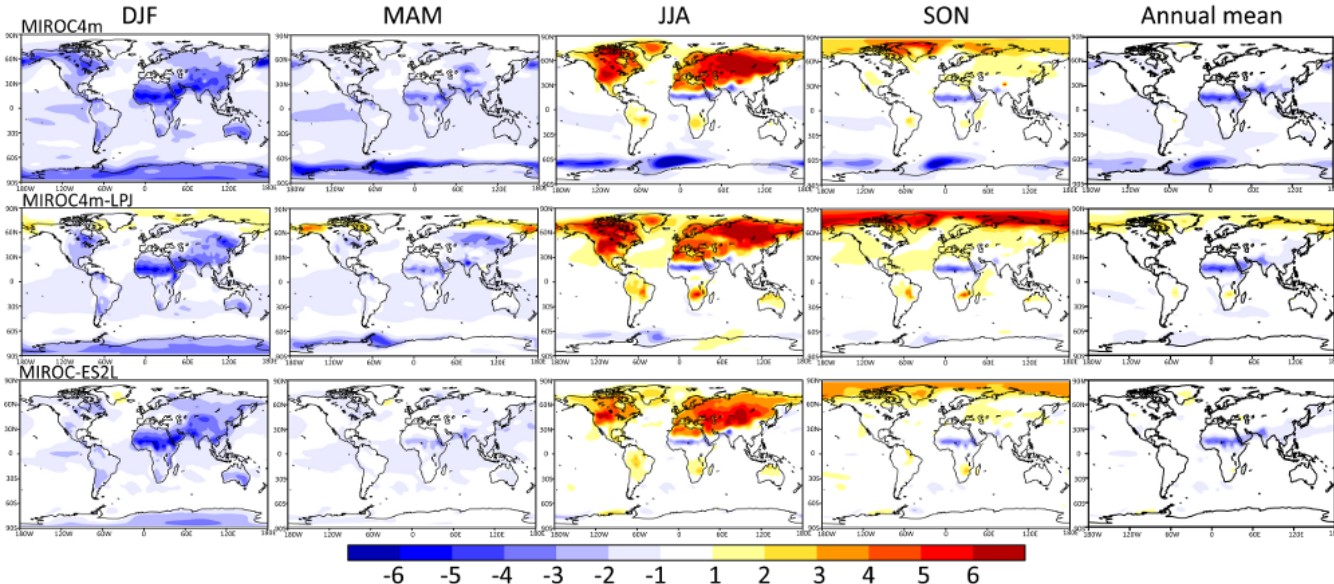

Figure 4: Seasonal and annual surface air temperature difference (K) between 127k and PI in three models. Calendar in 127k is adjusted by a method based on Bartlein and Shafer (2019).

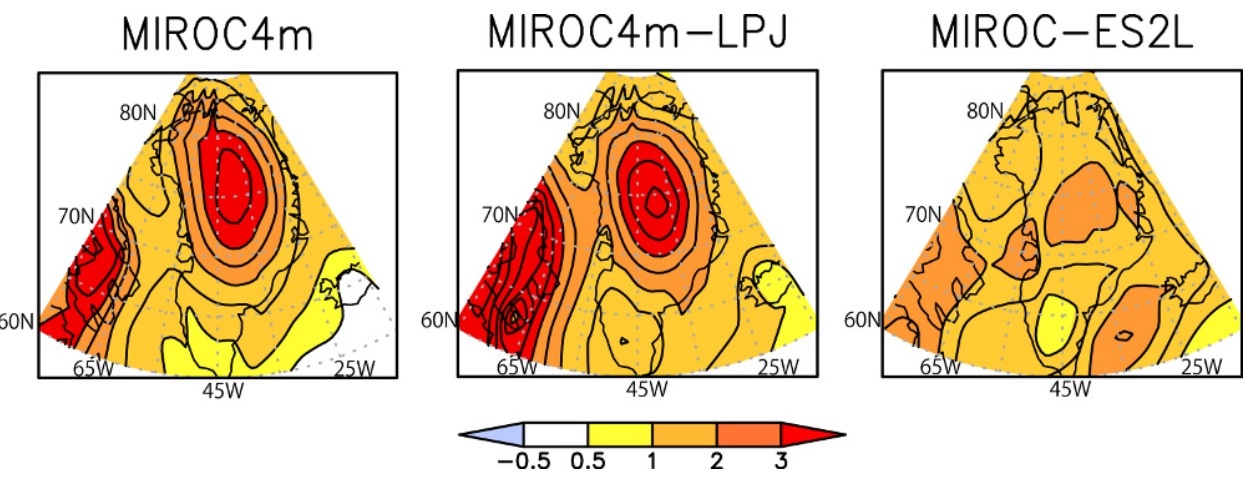

Figure 5: May-June-July-August averaged temperature difference (K) between 127k and PI in three models. Calendar in 127k is adjusted by a method based on Bartlein and Shafer (2019).

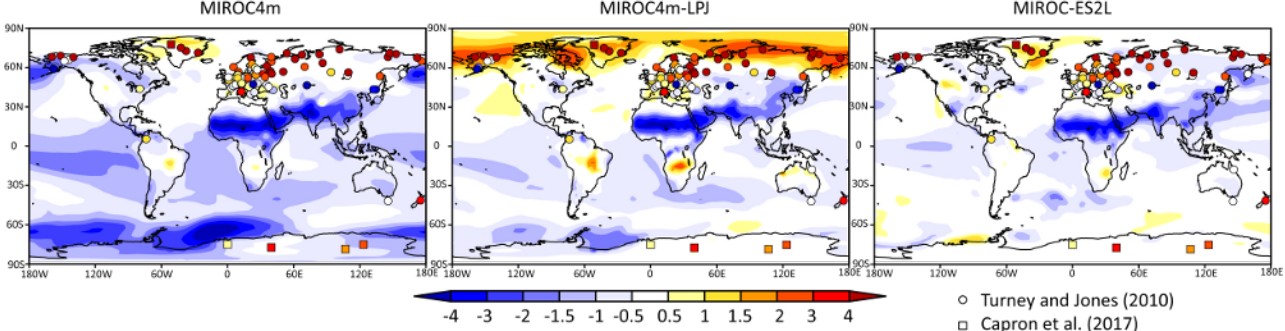

Figure 6: Annual surface air temperature change (K) between 127k and PI is compared with reconstruction by Turney and Jones (2010) and Capron et al. (2017).

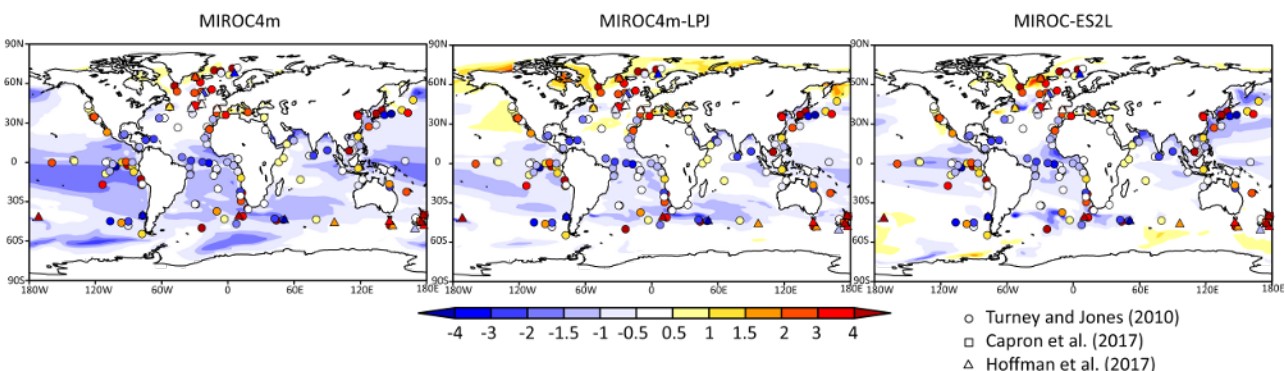

Figure 7: Annual sea surface temperature change (K) in 127k from PI is compared with reconstruction by Turney and Jones (2010), Capron et al. (2017) and Hoffman et al. (2017).

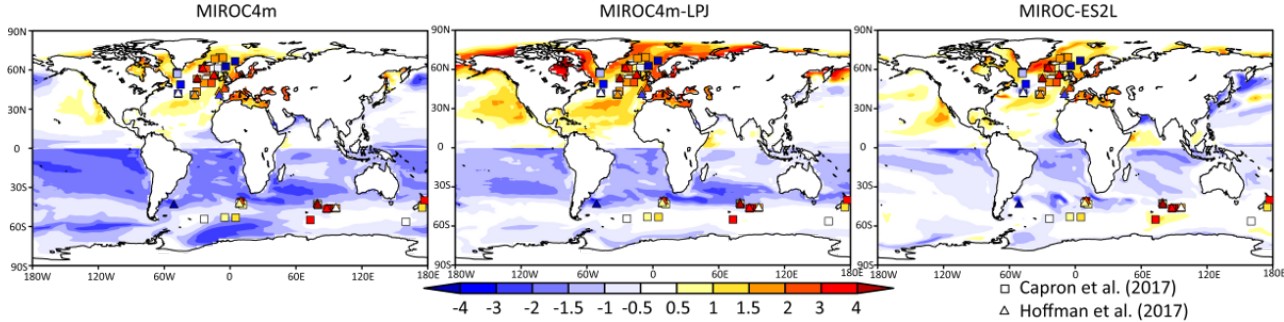

Figure 8: Summer (JJA in NH, DJF in SH) sea surface temperature change (K) in 127k from PI is compared with reconstruction by Capron et al. (2017) and Hoffman et al. (2017). Calendar in 127k is adjusted by a method based on Bartlein and Shafer (2019).

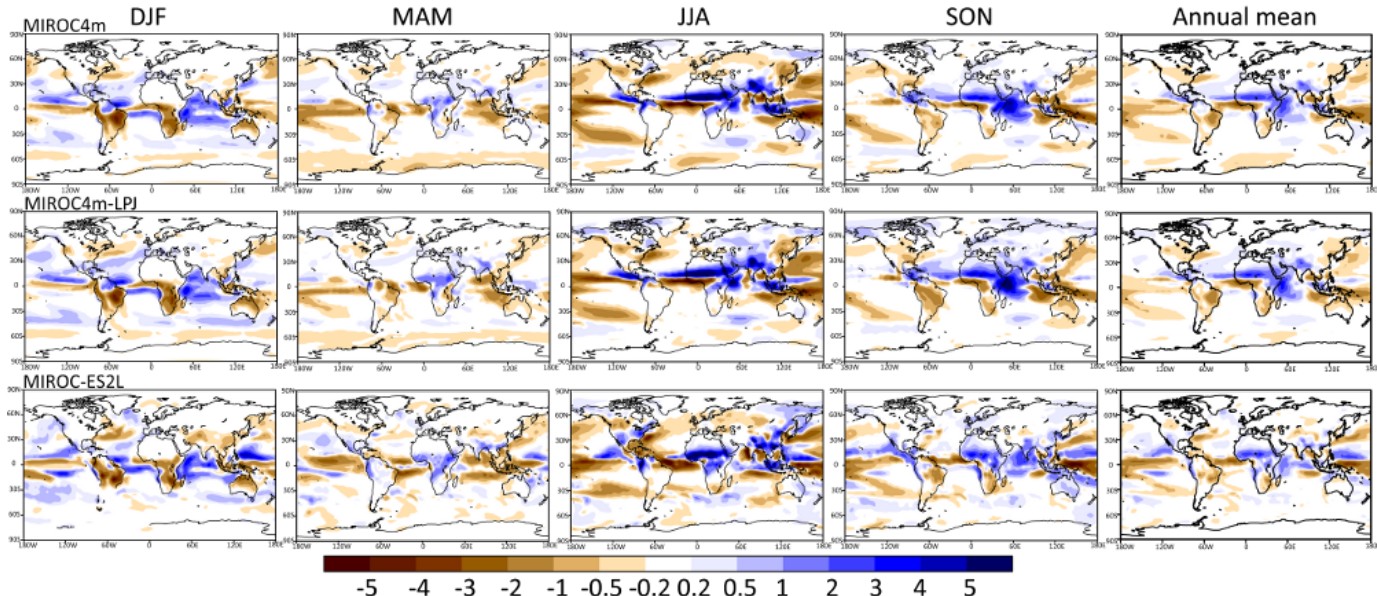

Figure 9: Annual and seasonal precipitation change (mm/day) is shown as the 127k-P. Calendar in 127k is adjusted by a method based on Bartlein and Shafer (2019).


# March sea ice concentration (%)

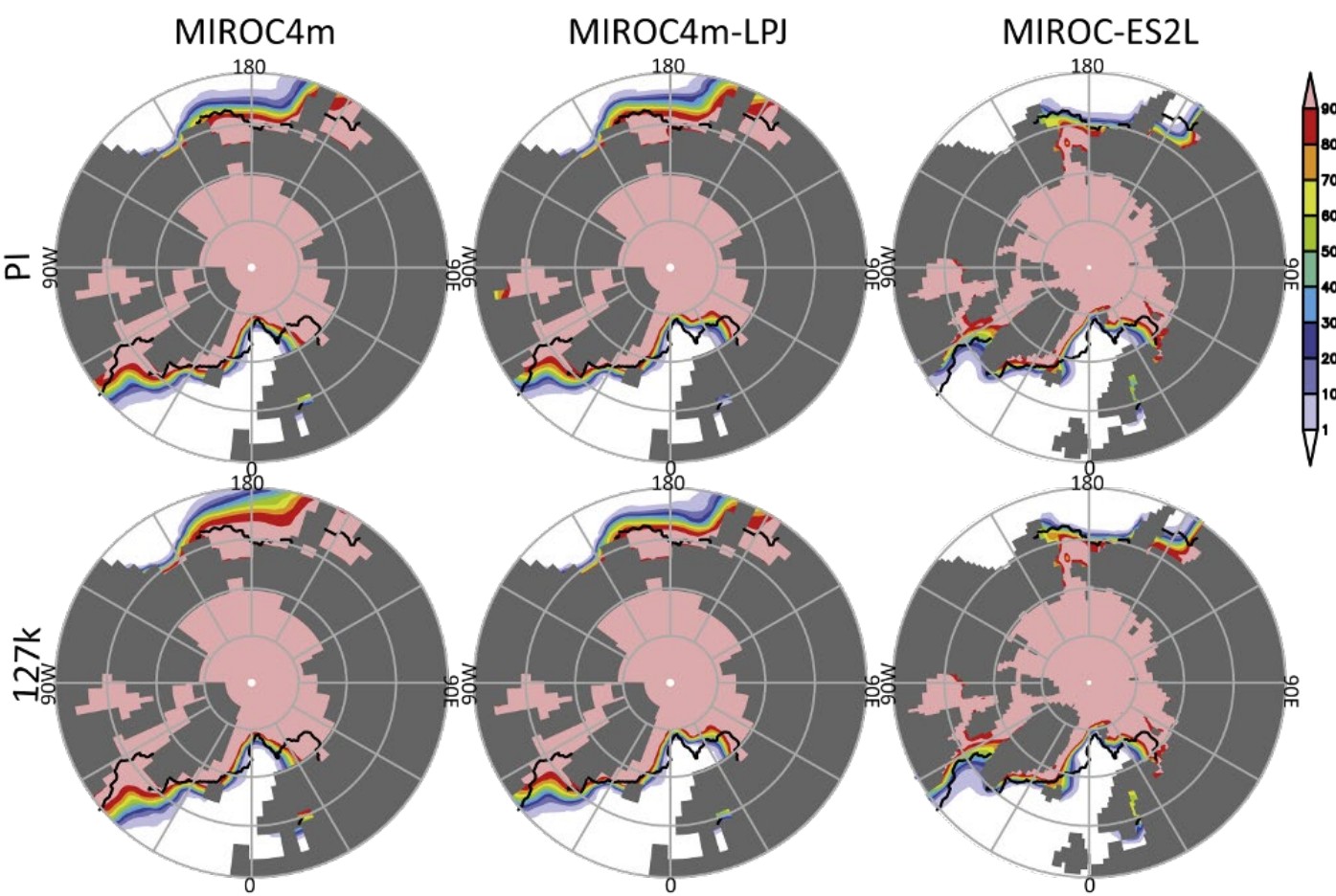

Figure 10: NH March sea ice concentration (%) in PI, 127k and difference between 127k and PI are shown for three models. Thick lines indicate a concentration of 15% in the climatology (the HadISST data averaged over 1870–1919). Calendar in 127k is adjusted by a method based on Bartlein and Shafer (2019).

# September sea ice concentration (%)

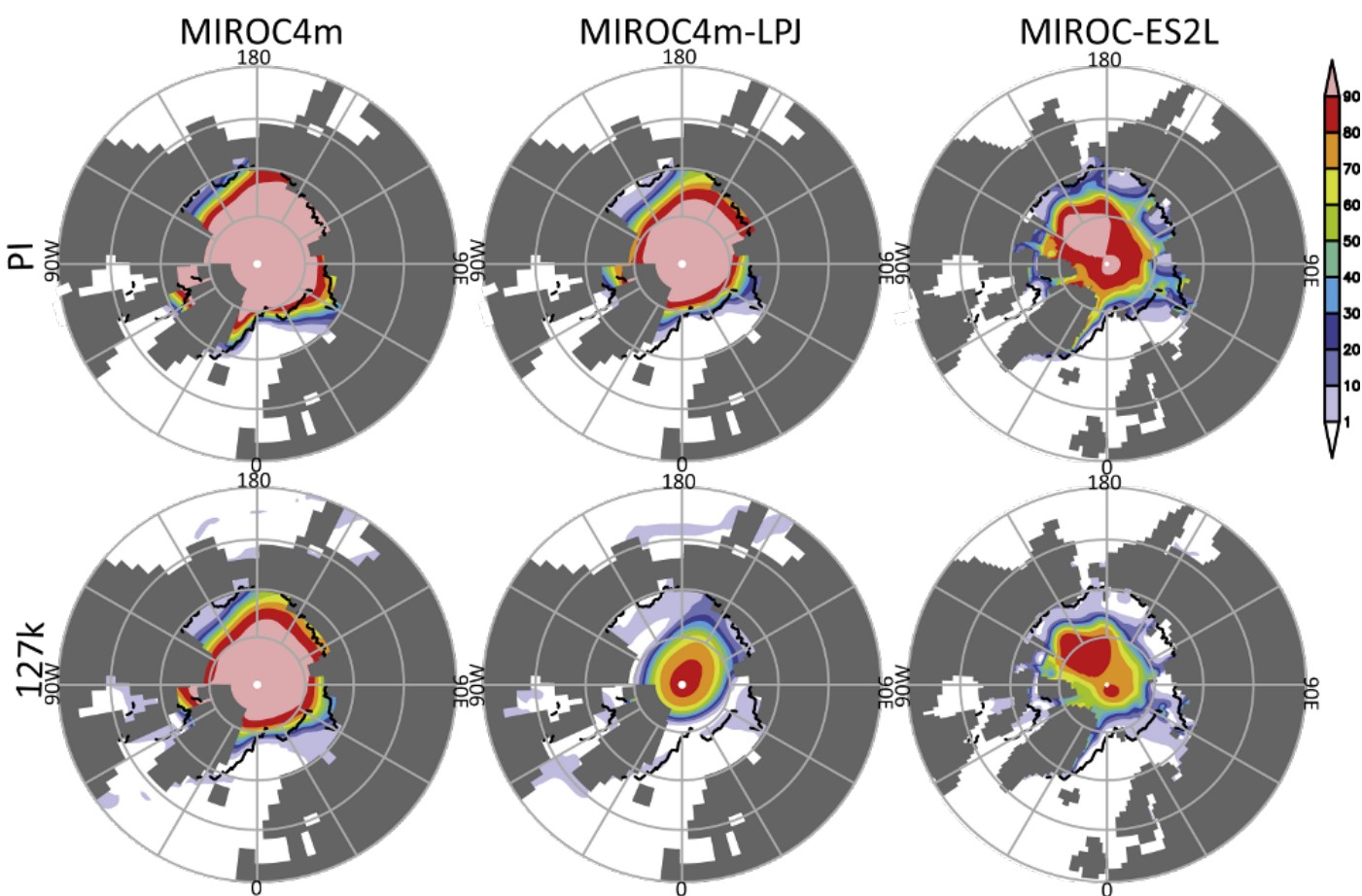

Figure 11: As same as Figure 10 but NH September sea ice concentration (%)

# March sea ice concentration (%)

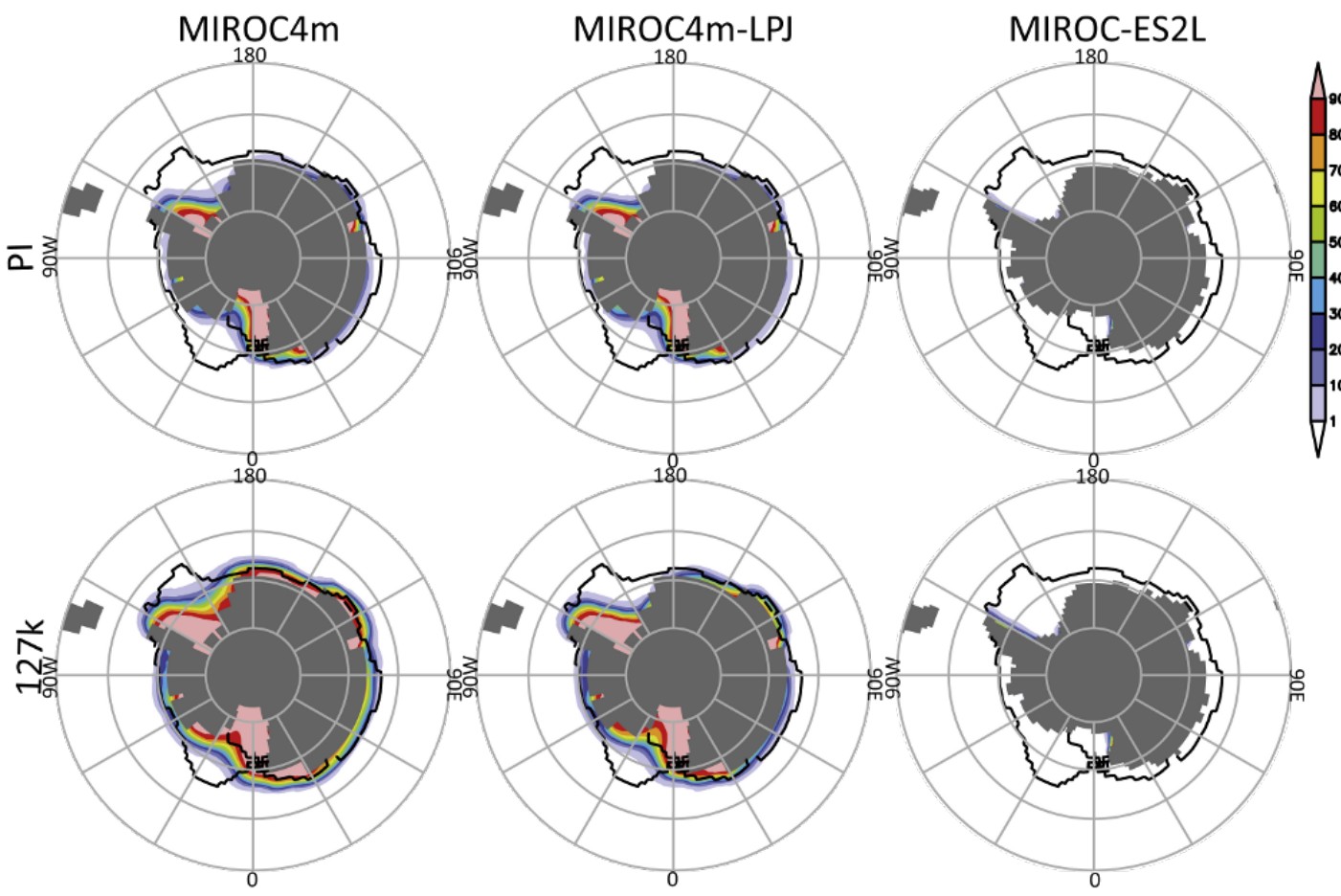

Figure 12: As same as Figure 10 but SH March sea ice concentration (%).


# September sea ice concentration (%)

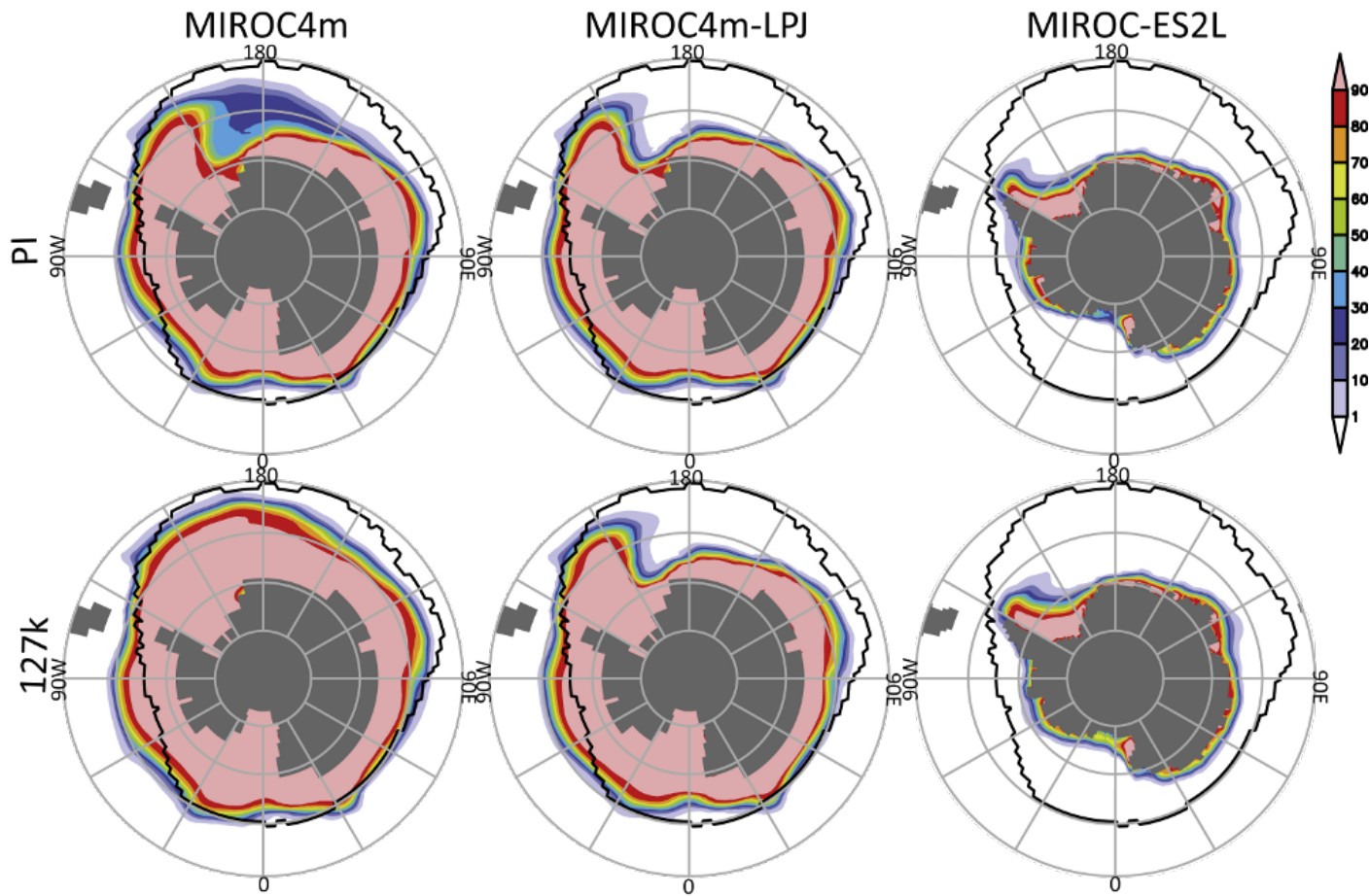

Figure 13: As same as Figure 10 but SH September sea ice concentration (%).


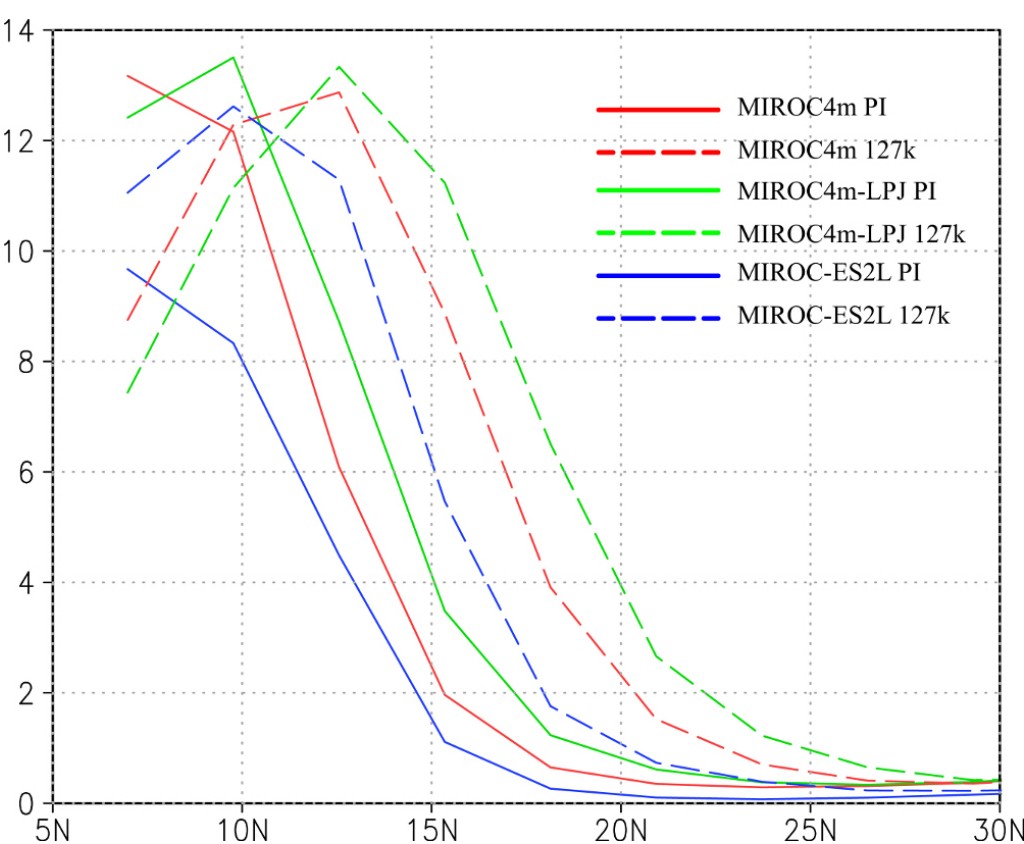

Figure 14: Zonally (30°W-20°E) averaged JJA precipitation (mm/day) over land at 5°N-30°N. Calendar in 127k is adjusted by a method based on Bartlein and Shafer (2019).


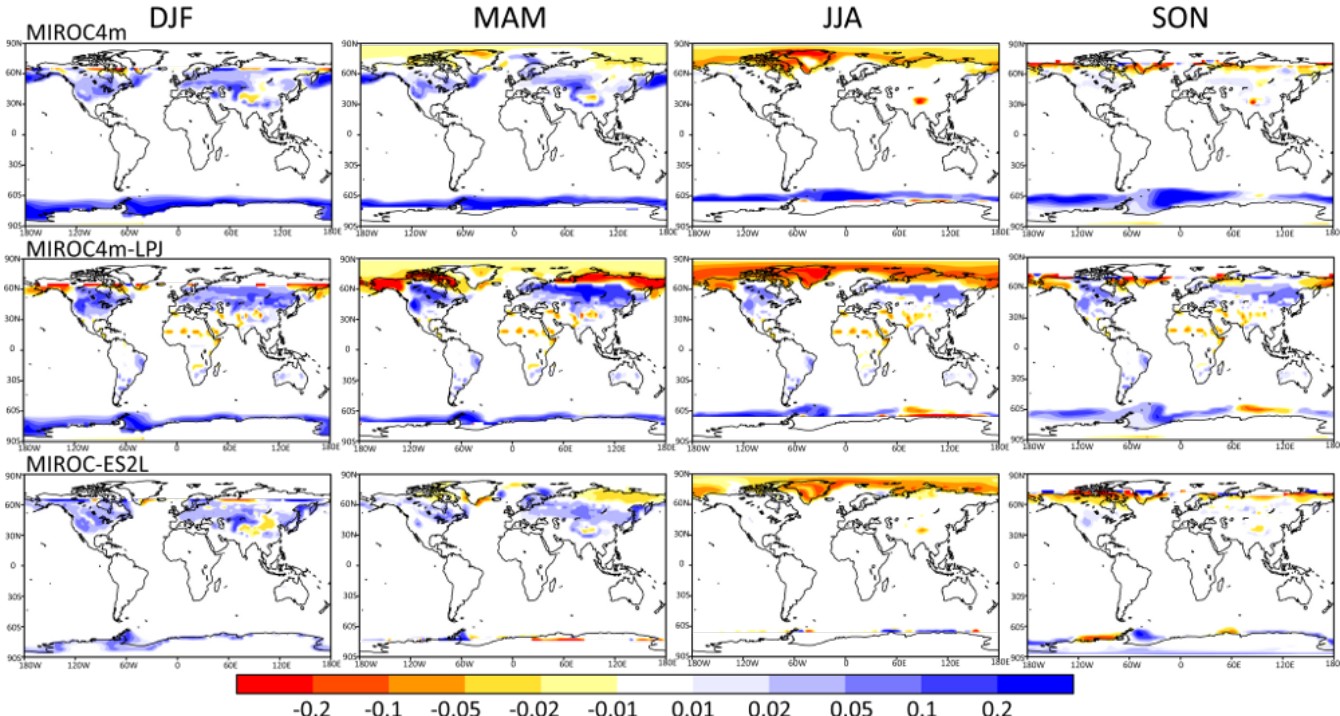

Figure 15: Seasonal surface albedo difference between 127k and PI in three models. Calendar in 127k is adjusted by a method based on Bartlein and Shafer (2019).

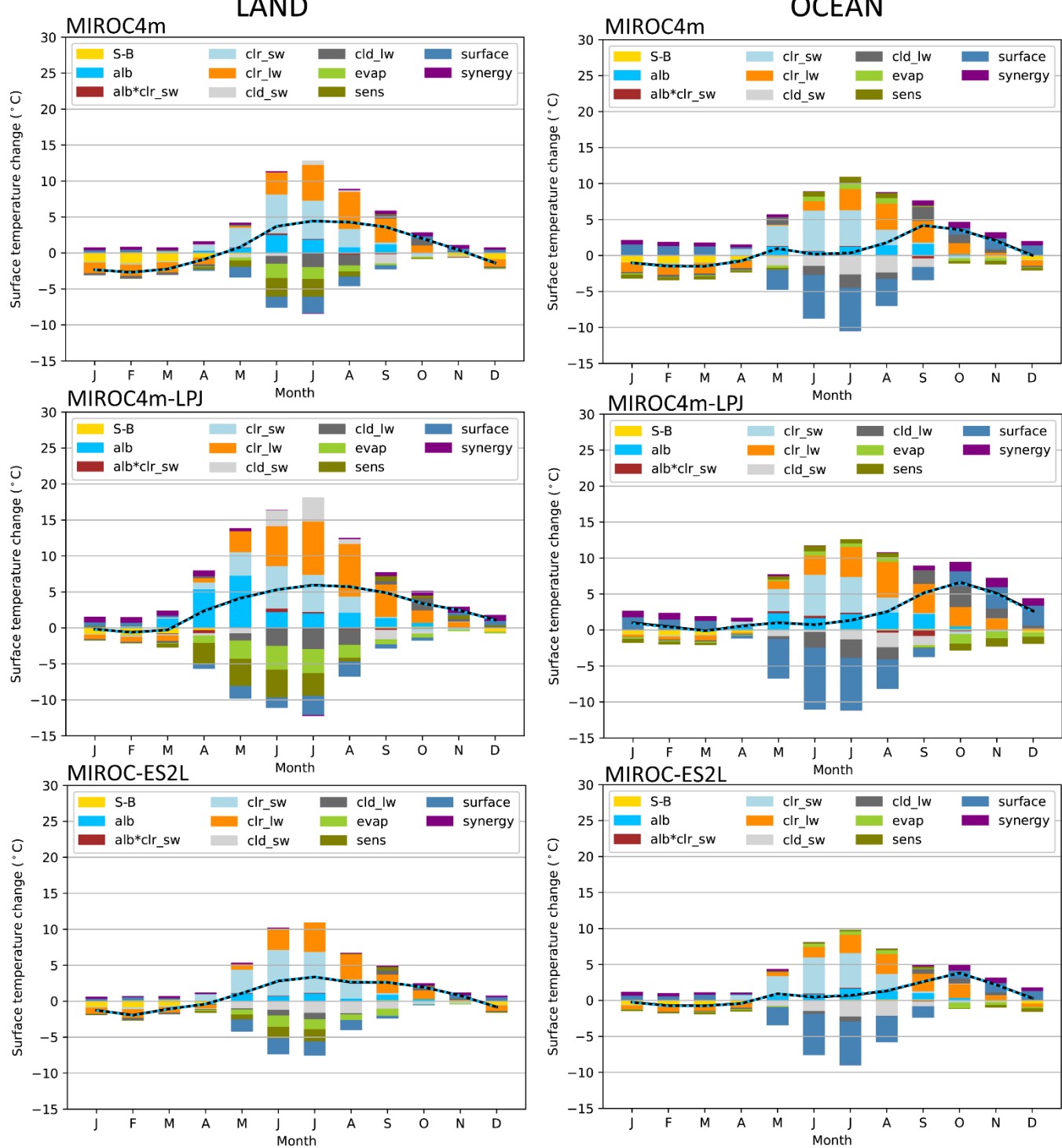


Figure 16: Simulated and diagnosed surface temperature changes (K) in 127k from PI for the land and the ocean (north of 60N). The solid black polygonal lines denote simulated changes and dashed blue lines denote the sum of the diagnosed partial changes; the two lines are superimposed. See Table 3 for the interpretation of each component.