# Peer review of "PMIP4/CMIP6 Last Interglacial simulations using three different versions of MIROC: importance of vegetation"

_Climate of the Past, 2019_

## Referee Comment (RC1) · Chuncheng Guo (Referee) · 2 Feb 2020

R. O'ishi and co-authors presented results of the CMIP6/PMIP4 lig127k simulation with three different versions of MIROC. They assessed large scale features of surface temperature, precipitation, and sea ice distributions in the simulated LIG climate, with extensive comparison to proxy dataset available. In addition, they addressed the importance of including vegetation feedbacks in getting Arctic warming at the northern high latitude, as indicated by proxy data.

I am overall positive about the manuscript. It fits the scope of the journal and the targeted special issue, and would serve as a useful reference for the audience interested

in this topic. However, I do have a number of comments on the manuscript as listed below (most of them are relatively minor), and hope the authors could address them before the manuscript can be considered for potential publication in Climate of the Past.

——————————

> The title is a bit misleading, e.g. ", with and without vegetation feedback" can read as sensitivity experiments for each version of the model. I do understand that the authors would like to address the importance of vegetation feedback, however, the authors could consider to drop this, as the majority of the paper is focused on the large scale features. In addition, it would be useful to highlight "using three different versions of MIROC" in the title.

> L8: It should be three LIG experiments; please rephrase.

> L12-13: too many commas in the sentence; suggest to change to, for example, ". . . vegetation distribution, shows annual mean warming signals at northern high latitudes, as indicated by proxy data."

> L27: For context please give a number/range of the estimated sea level rise for the LIG from the literature – Dutton et al. (2015) for example. Dutton et al., 2015, Sea-level rise due to polar ice-sheet mass loss during past warm periods, Science 349, aaa4019

> L21-29: I don't think the authors have presented sufficient background introduction for the LIG period. Such information might be obvious to certain experts but not so to the general audience. Could the authors further elaborate on the characteristics of the LIG climate, for example, on the sea level (see previous comment) and surface temperature? Especially, the authors have discussed extensively in the main text on the proxy-reconstructed temperature from different dataset, and some introduction here would be beneficial. In addition, a recent paper on the LIG precipitation (Scussolini et al., 2019) would also be helpful for the introduction here.

Scussolini et al., 2019, Agreement between reconstructed and modeled boreal precipitation of the Last Interglacial, Science Advances.

> L31; the three Braconnot papers should be combined into, e.g. Braconnot et al. (2000, 2007, 2012). The most recent PMIP4 overarching paper by Kageyama et al. (2018) should also be cited here.

> after L36: please expand with how the manuscript is structured.

> L40: "The AOGCM MIROC4m, is based on. . ."?

> L49: here and elsewhere; is there any reference for the sea ice model?

> L49: "These models are used . . ."; you mean MIROC4m, or the different components of it?

> L64: The model resolutions of (?) are the same as those of MIROC4m.

> L67-68: The reference of Hajima et al. appears in GMDD and should be updated here.

> L83: Otto-Bliesner et al. (2017)

> L83: piControl should be italic here.

> L85-86: ". . . using MIROC-ES2L to MIROC-ES2L." is strange. Please rephrase.

> L91: Figure 2 is not properly referred to in the text; I think it should come early in Section 2.1.

> L93: Although information can be found in Table 2, the authors should state the length of their simulations here – which is impressive by the way, and could be useful for the audience who are potentially interested in studies of equilibration and variability (multi-decadal/centennial) during the LIG.

> L98: please address that ">6 K" is only a regional feature.

> L104: please give global mean values of LIG annual mean temperature anomalies

relative to PI. This is important information for comparing with other model simulations, and should also be highlighted in the abstract.

> L112-114: It is up to the authors to decide, but I don't see the added value for including this paragraph here. Also, why MJJA is used here (rather than JJA)?

> L116: To my (very limited) knowledge on paleoclimate proxy dataset, the Turney and Jones (2010) dataset gives geographically asynchronous warming information, e.g. the compilation presents peak warmth information during the LIG rather than a time slice; see discussions by Capron et al. (2014, 2017). I therefore wonder the meaning and validity of making such a model-data comparison. The authors should take this into account, and at least make clear of the limit of the dataset and hence the comparison in the text.

> Figs 6&7: please consider merging these two figures. I understand that the core locations could be overlapping each other for the two datasets, but this could be avoided by, for example, putting them side to side and mention it in the figure caption.

> L120: "+1 K"; please make consistent of Celsius or Kelvin throughout the text.

> L126: "…at low latitudes"; you mean southern latitudes?

> L131, 133: "…warming in the northern Atlantic Ocean"; I guess the authors refer to comparison with that one single site in the Irminger Sea? If yes I don't think the comparison should be generalized to the "northern Atlantic Ocean".

> Section 3.2.1: The description on the LIG precipitation is inadequate. Please add more details on the main features/changes. The work by Scussolini et al. (2019) could be referred for comparison here. In addition, I wonder if it is more conventional to have an anomaly map (LIG minus PI) rather than the ratio map? I am not an expert on this, and it is up to the authors to decide.

> L149: "MIROC4m-LPJ" and "MIROC4m" should be swapped in this line?

> L154: "Figure 13"

> L168: It would be useful to overlie the observed sea ice extent in the figure.

> Figure 13: 1) please consider presenting the sea ice thickness in meters rather than centimeters (this also applies to other sea ice figures); 2) please change the colormap, as the current one is quite saturated with high values, and it is difficult to see the intervals (this may also apply to the other sea ice figures); 3) please consider overlying the observed (in the PI panels) and modeled (in the PI and 127k panels) 15% contour of sea ice extent (this also applies to the other sea ice figures). The field of sea ice thickness only is insufficient in presenting the sea ice distribution; this is especially the case for the SH sea ice extent.

> Figures 15&16: please change the range of the plotted sea ice thickness (and perhaps also colormap; see previous comment), e.g. it does not need to be the same with that of the NH.

> L172: "as well as" -> ", similar to"?

> L175: "seasons"

> L190: move "by +3K" to later in the sentence (after "North America")?

> L197: I would say "similar" rather than "basically the same".

> L209: "and in future"; I would say that such results have implications for future simulations.

> L210-211: I don't see the direct connection with climate sensitivity here. The LIG climate change relative to present day is related to differences in the orbital forcing rather than $CO_2$.

> L215: is there really a "warm bias" in MIROC4m-LPJ, or is it just warmer in this model version compared to the other two? If it is the former case, then this should be brought up earlier in the main text.

---

## Referee Comment (RC2) · Anonymous Referee #2 · 14 Feb 2020

The authors have run LIG simulations with three different versions of MIRCO model, the climate change between the LIG and PI are compared with proxy data. They show that all three models have produced expected warming in summer correspond to the imposed orbital forcing. Among the three models, only one model that is coupled with a dynamical vegetation model, is able to reproduce the annual warming over the northern high latitudes. And they conclude that vegetation feedback is necessary for reproducing the comparable warming showing in proxy data. The paper is an overall evaluation on simulated LIG climate by MIRCO, it is a good documentation for the LIG simulations from this specific model.The authors may consider the following comments and improve the manuscript.

1. The models are presented in a clear hierarchy, from an AOGCM, to a AOGCM coupled with LPJ vegetation and finally an earth system model with more physical processes are included. The results are obviously not following the complexity of the model components, instead it turned out the version OAGCM+LPJ version is the one that better agree with temperature reconstruction. Besides concluding that vegetation feedback play important role to simulate LIG climate, the authors should discuss more why the other components such as biogeochemistry over the land and in the ocean do not contribute much to the expected LIG warming. Does it imply these components are not important or these components do not work well in LIG climate?

2. For the major conclusion, the important contribution from vegetation feedback to reproduce the warming over land and northern high latitudes, the authors did not present the relevant feedbacks. Is it due to the change in the albedo, or LAI, or evaporations? According to the results from ESM, it seems LAI does not contribute much. It is not clear which physical processes are associated to LAI, if these processes do not contribute to the warming, does it mean they are not important at all or they are not well represented in this model? More discussions on these key questions would provide useful information on how to improve the model.

3. The presentation on the results are lack of the motivation, each section simply starts with showing the figures on temperature, precipitation, sea-ice and vegetation and did not mention why do authors show these results. We may understand that simulated temperature can be compared with proxy data and tell us if the warming in certain regions are well reproduced. What do the resulted precipitation and sea ice imply? It would be helpful that author can provide a few motivation sentences for their results presentation.

4. For the sea-ice, the authors may aim to show that sea-ice feedback is contributing to the high-latitude warming. In this case it is better to use the sea-ice extent instead of thickness, which can better explain the sea-ice-albedo feedback. It would be helpful to add some discussion on sea-ice feedback, instead of only showing the differences

in sea-ice.

5. For the model-data comparison, the temperature reconstructions from two datasets are used, but no any comments on the uncertainty of each datasets. I suggest the authors provide some information on the data uncertainties in order to gain fair evaluation for the model results.

6. Fig1 shows much less insolation in SON in northern high latitudes, but in all three models simulated warm Arctic in surface air temperature in fig4, especially a strong warming in MIROC4m-LPJ version. Authors should mention this in the text and give some explanations.

7. L129, "We also compare model annual SST..", should be "We also compare modelled annual SST...".

8. Line 150, this section is talking about the precipitation and it is strange to end with a sentence with vegetation distribution in Sahara. Consider to remove. It is more appropriate to mention it as in L193-194, so called "green Sahara" needs the reference, and specify how green it is? Do the proxy data indicate any specific vegetation type development in Sahara, and within how large area?

9. L161, "...on the eastern coast of Geendland", I observed the northern part of Greenland in figure 13. In fig13, it doesn't make sense that warm LIG climate produces more sea-ice than PI in March (fig1 shows more insolation in MAM in northern high latitudes), any explanation?

10. L171-172, "... as well as NH", this is confusing and suggest to rephrase.

Figures:

1. In Fig3, the Greenland ice-sheet distribution in ES2l looks different from others, any explanation?

2. Fig5, though it shows Greenland, would be helpful to have the lat-lon labels.
3. Fig6 and Fig7, it is difficult to distinguish the dark red and dark blue for proxy data, for example those over the Greenland, suggest to change the colour bar for a clear distinguish.

---

## Author Comment (AC1) · 26 May 2020

**To reviewer 1 (Dr. Chuncheng Guo)**

R. O'ishi and co-authors presented results of the CMIP6/PMIP4 lig127k simulation with three different versions of MIROC. They assessed large scale features of surface temperature, precipitation, and sea ice distributions in the simulated LIG climate, with extensive comparison to proxy dataset available. In addition, they addressed the importance of including vegetation feedbacks in getting Arctic warming at the northern high latitude, as indicated by proxy data.

I am overall positive about the manuscript. It fits the scope of the journal and the targeted special issue, and would serve as a useful reference for the audience interested

in this topic. However, I do have a number of comments on the manuscript as listed below (most of them are relatively minor), and hope the authors could address them before the manuscript can be considered for potential publication in Climate of the Past.

**We greatly appreciate that you agreed to review this paper and we thank you for your positive comments. We revised our manuscript following your and another reviewer's comments. Please find following response to specific comments and revised manuscript attached as a supplement.**

> The title is a bit misleading, e.g. ", with and without vegetation feedback" can read as sensitivity experiments for each version of the model. I do understand that the authors would like to address the importance of vegetation feedback, however, the authors could consider to drop this, as the majority of the paper is focused on the large scale features. In addition, it would be useful to highlight "using three different versions of MIROC" in the title.

**Corrected as suggested.**

> L8: It should be three LIG experiments; please rephrase.

**Corrected.**

> L12-13: too many commas in the sentence; suggest to change to, for example, "... vegetation distribution, shows annual mean warming signals at northern high latitudes, as indicated by proxy data."

**Corrected.**

> L27: For context please give a number/range of the estimated sea level rise for the LIG from the literature – Dutton et al. (2015) for example. Dutton et al., 2015, Sea-level rise due to polar ice-sheet mass loss during past warm periods, Science 349, aaa4019

**We will add the range of sea-level change, referring Dutton et al. (2015).**

> L21-29: I don't think the authors have presented sufficient background introduction for the LIG period. Such information might be obvious to certain experts but not so to the general audience. Could the authors further elaborate on the characteristics of the LIG climate, for example, on the sea level (see previous comment) and surface temperature? Especially, the authors have discussed extensively in the main text on the proxy-reconstructed temperature from different dataset, and some introduction here would be beneficial. In addition, a recent paper on the LIG precipitation (Scussolini et al., 2019) would also be helpful for the introduction here. Scussolini et al., 2019, Agreement between reconstructed and modeled boreal precipitation of the Last Interglacial, Science Advances.

**We added general quantitative explanations to show the overview of the LIG in Introduction with some more references.**

> L31; the three Braconnot papers should be combined into, e.g. Braconnot et al. (2000, 2007, 2012). The most recent PMIP4 overarching paper by Kageyama et al. (2018) should also be cited here.

**Corrected.**

> after L36: please expand with how the manuscript is structured.

**Corrected.**

> L40: "The AOGCM MIROC4m, is based on..."?

**Corrected.**

> L49: here and elsewhere; is there any reference for the sea ice model?

**Hasumi and Emori (2004: MIROC4m and -LPJ) and Komuro and Suzuki (2013: MIROC-ES2L) are referred.**

> L49: "These models are used ..."; you mean MIROC4m, or the different components of it?

**Corrected to MIROC4m.**

> L64: The model resolutions of (?) are the same as those of MIROC4m.

**The resolution part is corrected.**

> L67-68: The reference of Hajima et al. appears in GMDD and should be updated here.

**Corrected.**

> L83: Otto-Bliesner et al. (2017)

**Corrected.**

> L83: piControl should be italic here.

**Corrected.**

> L85-86: "... using MIROC-ES2L to MIROC-ES2L." is strange. Please rephrase.

**Corrected.**

> L91: Figure 2 is not properly referred to in the text; I think it should come early in Section 2.1.

**Figure 2 is referred in the beginning of Section 2.1**

> L93: Although information can be found in Table 2, the authors should state the length of their simulations here – which is impressive by the way, and could be useful for the audience who are potentially interested in studies of equilibration and variability (multi-decadal/centennial) during the LIG.

**Simulation length is mentioned in the text.**

> L98: please address that ">6 K" is only a regional feature.

**Corrected.**

> L104: please give global mean values of LIG annual mean temperature anomalies relative to PI. This is important information for comparing with other model simulations, and should also be highlighted in the abstract.

**Global averaged temperature change is rather a cooling in the present study. These values, -0.94K(MIROC4m), -0.39K(MIROC4m-LPJ) and -0.43(MIROC-ES2L), will be mentioned and discussed in the text.**

> L112-114: It is up to the authors to decide, but I don't see the added value for including this paragraph here. Also, why MJJA is used here (rather than JJA)?

**We add a motivation of MJJA averaged figures.**

> L116: To my (very limited) knowledge on paleoclimate proxy dataset, the Turney and Jones (2010) dataset gives geographically asynchronous warming information, e.g. the compilation presents peak warmth information during the LIG rather than a time slice; see discussions by Capron et al. (2014, 2017). I therefore wonder the meaning and validity of making such a model-data comparison. The

authors should take this into account, and at least make clear of the limit of the dataset and hence the comparison in the text.

**We explain Turney and Jones (2010) indicate the maximum change during the whole LIG and not directly comparable to 127k model result. However, sign and pattern is worth comparing with model result since number of data site is very small in newer transient-based reconstructions (Capron et al., Hoffman et al. 2017).**

> Figs 6&7: please consider merging these two figures. I understand that the core locations could be overlapping each other for the two datasets, but this could be avoided by, for example, putting them side to side and mention it in the figure caption.

**Figures 6 and 7 will be corrected along both reviewer's comments.**

> L120: "+1 K"; please make consistent of Celsius or Kelvin throughout the text.

**Corrected**

> L126: "…at low latitudes"; you mean southern latitudes?

**Corrected to "tropical area".**

> L131, 133: "…warming in the northern Atlantic Ocean"; I guess the authors refer to comparison with that one single site in the Irminger Sea? If yes I don't think the comparison should be generalized to the "northern Atlantic Ocean".

**Corrected to "the Irminger Sea".**

> Section 3.2.1: The description on the LIG precipitation is inadequate. Please add more details on the main features/changes. The work by Scussolini et al. (2019) could be referred for comparison here. In addition, I wonder if it is more conventional to have an anomaly map (LIG minus PI) rather than the ratio map? I am not an expert on this, and it is up to the authors to decide.

**We explained precipitation more detailed and qualitative. Precipitation change is drawn as difference. We also compared results with Scussolini et al. (2019).**

> L149: "MIROC4m-LPJ" and "MIROC4m" should be swapped in this line?

**Last sentence is removed.**

> L154: "Figure 13"

**Corrected.**

> L168: It would be useful to overlie the observed sea ice extent in the figure.

**Modified as suggested.**

> Figure 13: 1) please consider presenting the sea ice thickness in meters rather than centimeters (this also applies to other sea ice figures); 2) please change the colormap, as the current one is quite saturated with high values, and it is difficult to see the intervals (this may also apply to the other sea ice figures); 3) please consider overlying the observed (in the PI panels) and modeled (in the PI and 127k panels) 15% contour of sea ice extent (this also applies to the other sea ice figures). The field of sea ice thickness only is insufficient in presenting the sea ice distribution; this is especially the case for the SH sea ice extent.

**Modified as suggested.**

> Figures 15&16: please change the range of the plotted sea ice thickness (and perhaps also colormap; see previous comment), e.g. it does not need to be the same with that of the NH.

**Modified as suggested.**

> L172: "as well as" -> ", similar to"?

**Corrected.**

> L175: "seasons"

**Corrected.**

> L190: move "by +3K" to later in the sentence (after "North America")?

**Corrected.**

> L197: I would say "similar" rather than "basically the same".

**Corrected.**

> L209: "and in future"; I would say that such results have implications for future simulations.

**This part is rephrased.**

> L210-211: I don't see the direct connection with climate sensitivity here. The LIG climate change relative to present day is related to differences in the orbital forcing rather than CO2.

**This part is removed.**

> L215: is there really a "warm bias" in MIROC4m-LPJ, or is it just warmer in this model version

compared to the other two? If it is the former case, then this should be brought up earlier in the main text.

**We removed the word "bias" and just noticed that different PI states lead to different responses in warm climate. We may add comparison with observed values in supplementary material later.**

---

## Author Comment (AC2) · 26 May 2020

**To reviewer 2**

The authors have run LIG simulations with three different versions of MIRCO model, the climate change between the LIG and PI are compared with proxy data. They show that all three models have produced expected warming in summer correspond to the imposed orbital forcing. Among the three models, only one model that is coupled with a dynamical vegetation model, is able to reproduce the annual warming over the northern high latitudes. And they conclude that vegetation feedback is necessary for reproducing the comparable warming showing in proxy data. The paper is an overall evaluation on simulated LIG climate by MIRCO, it is a good documentation for the LIG simulations from this specific model. The authors may consider the following comments and improve the manuscript.

**We greatly appreciate that you agreed to review this paper and we thank you for your positive comments. We revised our manuscript following your and another reviewer's comments. Please find following response to specific comments and revised manuscript attached as a supplement.**

1. The models are presented in a clear hierarchy, from an AOGCM, to a AOGCM coupled with LPJ vegetation and finally an earth system model with more physical processes are included. The results are obviously not following the complexity of the model components, instead it turned out the version OAGCM+LPJ version is the one that better agree with temperature reconstruction. Besides concluding that vegetation feedback play important role to simulate LIG climate, the authors should discuss more why the other components such as biogeochemistry over the land and in the ocean do not contribute much to the expected LIG warming. Does it imply these components are not important or these components do not work well in LIG climate?
**In all experiments, GHG values are fixed to specific values in all models including MIROC-ES2L along PMIP4 protocol. It means that land and ocean ecosystem do not affect on atmospheric GHGs. The different GHGs between PI and 127k may cause slight difference on LAI through difference of photosynthesis, but it is negligible. We add explanation of fixed GHG treatment in the experimental settings.**

2. For the major conclusion, the important contribution from vegetation feedback to reproduce the warming over land and northern high latitudes, the authors did not present the relevant feedbacks. Is it due to the change in the albedo, or LAI, or evaporations? According to the results from ESM, it seems LAI does not contribute much. It is not clear which physical processes are associated to LAI, if these processes do not contribute to the warming, does it mean they are not important at all or they are not well represented in this model? More discussions on these key questions would provide

useful information on how to improve the model.

**This comment is related to comments 4 and 6. We explain the feedback processes focusing on the surface energy balance by referring previous studies (Laîné et al. 2015 and Yoshimori and Suzuki 2019). To explain the effect of biogeophysical feedback on surface temperature, we will add figures focusing on change in albedo. This figure also helps explanations to the reviewer's comments 4 and 6. In MIROC4m-LPJ, not only LAI increase but also vegetation height reduces snow amount in MAM -> reduction of albedo larger than that of due to vegetation change in MAM -> larger ice reduction in the Arctic Ocean in summer -> heat stored in Arctic Ocean -> stored heat is emitted in SON. We plan to add discussion on changes in surface energy flux components and/or related variables over the largely warmed land region."**

3. The presentation on the results are lack of the motivation, each section simply starts with showing the figures on temperature, precipitation, sea-ice and vegetation and did not mention why do authors show these results. We may understand that simulated temperature can be compared with proxy data and tell us if the warming in certain regions are well reproduced. What do the resulted precipitation and sea ice imply? It would be helpful that author can provide a few motivation sentences for their results presentation.

**In the present study, we clarify how vegetation feedback affects on the LIG climate, throughout different processes in different seasons. Precipitation change is important to understand vegetation distribution and its feedback. Sea ice change is also an essential ingredient of warming amplification.**

4. For the sea-ice, the authors may aim to show that sea-ice feedback is contributing to the high-latitude warming. In this case it is better to use the sea-ice extent instead of thickness, which can better explain the sea-ice-albedo feedback. It would be helpful to add some discussion on sea-ice feedback, instead of only showing the differences in sea-ice.

**Sea ice figures will be replaced to sea ice extent (with observed value, suggested by reviewer #1). We will add explanations on the interseasonal effect of sea ice feedback referring previous feedback studies (Laîné et al. 2015, Yoshimori and Suzuki 2019) which also answers to comment 2 and 6. We plan to add discussion on changes in surface energy flux components and/or related variables over the Arctic ocean region.**

5. For the model-data comparison, the temperature reconstructions from two datasets are used, but no any comments on the uncertainty of each datasets. I suggest the authors provide some information on the data uncertainties in order to gain fair evaluation for the model results

**We explain Turney and Jones (2010) indicate the maximum change during the whole LIG and not**

**directly comparable to 127k model result. However, sign and pattern is worth comparing with model result since number of data site is very small in newer transient-based reconstructions (Capron et al., Hoffman et al. 2017).**

6. Fig1 shows much less insolation in SON in northern high latitudes, but in all three models simulated warm Arctic in surface air temperature in fig4, especially a strong warming in MIROC4m-LPJ version. Authors should mention this in the text and give some explanations.
**The explanation is related to the reply to reviewer's comment 2 and 4. Due to an intra-seasonal feedback, sea ice change in summer causes warming in SON in spite of less insolation in the Arctic Ocean in the LIG.    We will add detailed explanation in discussion part.**

7. L129, "We also compare model annual SST..", should be "We also compare modelled annual SST...".
**Corrected.**

8. Line 150, this section is talking about the precipitation and it is strange to end with a sentence with vegetation distribution in Sahara. Consider to remove. It is more appropriate to mention it as in L193-194, so called "green Sahara" needs the reference, and specify how green it is? Do the proxy data indicate any specific vegetation type development in Sahara, and within how large area?
**This part is removed as suggested. We referred Larrasoaña et al. (2013) as reference of Green Sahara in the LIG. This research estimated the expansion of vegetation in the LIG from correspondent wetness in the LIG.**

9. L161, "...on the eastern coast of Geendland", I observed the northern part of Greenland in figure 13. In fig13, it doesn't make sense that warm LIG climate produces more sea-ice than PI in March (fig1 shows more insolation in MAM in northern high latitudes), any explanation?
**Modified to "northern part of Greenland". The difference of sea-ice shows reduction in the major part of the Arctic Ocean in all three models, though small increase occurs in coastal region. As same as comment 4, we plan to add explanations based on surface energy fluxes.**

10. L171-172, ".... as well as NH", this is confusing and suggest to rephrase.
**Modified to "similar to".**

Figures:
1. In Fig3, the Greenland ice-sheet distribution in ES2l looks different from others, any explanation?
**MIROC-ES2L applies a new definition of land-sea mask than that of MIROC4m. The distribution of prescribed vegetation is also re-defined from newer satellite data set than MIROC4m. These**

**explanations are added to the model section.**

2. Fig5, though it shows Greenland, would be helpful to have the lat-lon labels.

**To be corrected.**

3. Fig6 and Fig7, it is difficult to distinguish the dark red and dark blue for proxy data, for example those over the Greenland, suggest to change the colour bar for a clear distinguish.

**Figures 6 and 7 are corrected along both reviewer's comments.**

---

## Author Response (AR2)

**To reviewer 2 (Qiong Zhang)**

The authors have made efforts to improve the manuscript by considering two reviewers' comments. **Thank you very much for your review and comments. Our manuscript is much more improved by introducing your suggestions.**

I have gone through the author response and revised manuscript and found some inconsistence. Major part appears in the sea ice part, where author followed my suggestion and presented the sec ice concentration in Fig 10-13, but did not change the presentation in the manuscript (section 3.2.2), where they still talk about the ice thickness.
**This was my mistake. Corresponding texts are modified.**

When show the sea-ice concentration, the change from 127K and PI can be visually observed, the difference between the two does not provide more helpful information but may lead to some confusion, such as those ocean area without sea-ice both in PI and 127K now appear in a yellow colour (0-10% in scaling). I suggest to remove the third panel of sea-ice difference between the two periods.
**We removed the difference of sea ice concentration in Figures 10, 11, 12 and 13.**

A few minor comments:
1. All the figures do not label latitudes and longitudes, expect Figure 5 which I asked to add lat-lon, it would be better to have lat-lon labels in all the figures.
**We added lon-lat labels in all figures as far as possible.**

2. In line 111, it is already defined that in this paper the two experiments refer to PI and 127K, but in table 1 and table 2 it still use piControl and lig127k, and in fig1, fig7, fig9 used 127ka, should be consistent throughout the paper.
**Name of experiments are unified throughout this paper.**

3. Line 126, "… referred to by MATSITO", does it mean "… referred to MATSITO"?
**This part is modified.**

4. Fig 6 caption, unit K should be after the temperature change, the same for the unit of precipitation in Fig 9.
**These units are relocated as suggested.**

5. Fig6 and Fig7, it is better to show all the proxy data in one figure. When it is separated, it is hard to compare the large scale feature

**Two proxy data are combined into one figure in Figures 6 and 7.**

**To Editor (Bette Otto-Bliesner)**

**Thank you for your suggestions. We introduced the calendar adjustment and feedback analysis method. New figures became more helpful in explaining our results.**

Please consider:

1) Applying the PaleoCalAdjust software to your monthly output. It is available at https://github.com/pjbartlein. As shown in Bartlein and Shafer, GMD, 2019, it is especially important for 127ka and seasonal anomalies. Particularly, your Figures 1, 4, 8, and 9.

**We applied PaleoCalAdjust to all monthly output. All related figures are redrawn (figures 1, 4, 5, 8, 9, 10, 11, 12, 13, 14 and 15). To keep consistency, corresponding texts are modified.**

2) Including one additional figure, the surface temperature changes for the Arctic (north of 60N) decomposed as in Fig. 5 of Yoshimori and Suzuki, 2019. M. Yoshimori is a co-author of this paper.

**We applied the same method as Yoshimori and Suzuki (2019) to our result. New texts, table and a figure are appended.**

3) The HadISST data averaged over 1870-1919 which is more appropriate for comparison to PI.

**HadISST sea ice is averaged over 1870-1919 in figures 10, 11, 12 and 13.**